# Velocity bias in intrusive gas-liquid flow measurements

B. Hohermuth [1]✉, M. Kramer [2], S. Felder [3] & D. Valero [4]

Gas–liquid flows occur in many natural environments such as breaking waves, river rapids and human-made systems, including nuclear reactors and water treatment or conveyance infrastructure. Such two-phase flows are commonly investigated using phase-detection intrusive probes, yielding velocities that are considered to be directly representative of bubble velocities. Using different state-of-the-art instruments and analysis algorithms, we show that bubble–probe interactions lead to an underestimation of the real bubble velocity due to surface tension. To overcome this velocity bias, a correction method is formulated based on a force balance on the bubble. The proposed methodology allows to assess the bubble–probe interaction bias for various types of gas-liquid flows and to recover the undisturbed real bubble velocity. We show that the velocity bias is strong in laboratory scale investigations and therefore may affect the extrapolation of results to full scale. The correction method increases the accuracy of bubble velocity estimations, thereby enabling a deeper understanding of fundamental gas-liquid flow processes.

[1] Laboratory of Hydraulics, Hydrology and Glaciology (VAW), ETH Zurich, Zurich, Switzerland. [2] School of Engineering and Information Technology (SEIT), UNSW Canberra, Campbell, Australia. [3] Water Research Laboratory, School of Civil and Environmental Engineering, UNSW Sydney, Sydney, Australia. [4] Water Resources and Ecosystems Department, IHE Delft Institute for Water Education, Delft, the Netherlands. ✉email: hohermuth@vaw.baug.ethz.ch

G as–liquid flows play an important role in mass, momentum and energy transfer. Some examples include rapids in mountain streams, breaking waves[1], nuclear reactors[2], process engineering plants[3] and violent flows in water conveyance infrastructures[4]. Mass and energy transport processes across a gas–liquid interface are dominated by shear and turbulence[5]. Accurate velocity measurements in gas–liquid flows are essential to properly evaluate gas–liquid interaction[6] and therefore lay the foundation for an improved modeling of gas–liquid flows. For example, air is entrained into the water body of an open channel flow if turbulent motion significantly distorts the free-surface[6–8], leading to entrainment of large air pockets that subsequently break up into bubbles of smaller diameters[1] ($0.1 \lesssim d \lesssim 100$ mm). When the time-averaged local void fraction ($C$) exceeds 3–5%, well-established mono-phase optical and acoustic flow measurement instrumentation is not able to measure the velocity field as the dispersed phase hinders the transmission of light and sound. Similarly, image analysis techniques cannot measure the internal flow structure beyond the sidewall or the free surface for moderate to high void fractions[9,10].

In previous works, internal properties of self-aerated flows have been widely measured with double-tip phase-detection intrusive probes[11,12], which have also been frequently used to characterize other gas–liquid flows[13–16]. The two needle tips of a double-tip probe are separated by a distance $\Delta x$ in probe-wise direction and changes in physical properties, such as electric resistance (conductivity probes, CP) or optical refraction (fiber-optical probes, FO), are synchronously sampled. Consequently, the arrival times of bubbles or droplets (i.e., particles) can be used to infer velocities. In highly aerated open-channel flows, the void fraction ranges from almost zero to unity and the most probable travel time of gas–liquid interfaces is typically obtained through a cross-correlation analysis, allowing the estimation of mean velocities, averaged over the sampling period. For such flow conditions, the necessary algorithms to extract pseudo-instantaneous velocity time series have been developed only recently[17,18].

An intrinsic limitation of phase-detection probes is the intrusiveness of the needle tips. Measured velocities may be subject to different velocity biases, which include (i) statistical velocity bias due to the fact that more particles impact the probe tips at high velocities[17,18], (ii) velocity bias due to the misalignment of the probe tips with flow streamlines[19] and (iii) velocity bias due to particle-probe interaction[20]. The statistical bias (i) can be corrected using appropriate weighting schemes as commonly used for other irregular sampling techniques such as laser Doppler anemometry (LDA)[18,21]. Even when the probe tips are aligned with the mean flow streamlines, the bias (ii) introduced by transverse velocity fluctuations is important for an accurate estimation of probe-wise velocity fluctuations and may be reduced using a robust filtering approach[18]. Ultimately, miniaturized multi-tip probes are needed to measure three-dimensional velocities and account for transverse and vertical velocity fluctuations[22]. Bias (iii) is due to interactions between dispersed-phase particles (bubbles/droplets) and the probe tips. Different mechanisms such as blinding, drifting and crawling have been recognized to affect the measured properties at low bubble Reynolds numbers ($Re_b \lesssim 10^2$)[20,23–26], while impact and crawling forces are anticipated to dominate in high Reynolds number flows ($Re_b \gtrsim 10^3$). However, generalized correction methods for bubble–probe interactions are missing. The blinding effect primarily leads to an underestimation of void fraction[24] and the effects of drifting are similar to the misalignment bias (ii). Therefore, we focus on the effects of impact and crawling, i.e. the deformation and deceleration along the probe when an air bubble impacts the probe tips as documented in high-speed videos[20,27].

Here, we introduce a fundamental description of bubble–probe interaction, which determines the velocity bias due to probe intrusiveness and recovers the undisturbed real bubble velocity from measured quantities. Accounting for the bubble–probe interaction effects, our correction scheme improved the velocity estimates of instantaneous velocities extracted from the probe tips' signals.

## Results

**Experiment**. We conducted experiments in a high-velocity air–water flow for three unit water discharges $q$ corresponding to bulk Reynolds numbers of $Re = q/v_c = [0.9 \cdot 10^6; 1.3 \cdot 10^6; 1.6 \cdot 10^6]$, where $v_c$ = kinematic continuous-phase (water) viscosity. Instantaneous dispersed-phase velocities ($u_d$) were measured with phase-detection intrusive probes and continuous-phase velocities ($u_c$) were recorded with LDA. To limit the distortion of the laser beams by air bubbles and to facilitate comparative velocity analyses with phase-detection probes, the measured profiles were located close to the sidewall at $z = 0.03$ m (Fig. 1b). Further details are presented in the Methods section and a detailed description of the experimental setup is given by Felder et al.[28].

**Macro- and microscopic flow properties**. The flow downstream of the inflow sluice gate (Fig. 1a) reached the highest flow velocities at the maximum flow contraction and decelerated further downstream. Air entrainment started shortly after the flow contraction due to strong turbulence at the air–water interface, creating a rapidly varied, fully aerated flow in the first third of the tunnel chute[28]. In the middle and in the end sections of the chute, the high-velocity air–water flows were gradually varied[29] and free of inlet effects, including the measurement location $x = 15.72$ m (Fig. 1c–e).

The time-averaged void fraction profile shown in Fig. 2a agreed with shapes typically observed in self-aerated open channel flows[30]. Void fractions were small ($C < 0.1$) up to a dimensionless elevation of $y^+ \approx 3 \cdot 10^4$, where $y^+ = yu^*/v_c$, with $u^*$ = shear velocity and $y$ = wall-normal coordinate. As shown later, the particle size is an important parameter to assess bubble–probe interactions. The Sauter diameter is used to characterize the particle size of bubbles and droplets[31–34]. Herein, we define the window-averaged Sauter diameter $d = 1.5u_d C_i/F_i$, where $C_i, F_i$ are the void fraction and particle frequency of the $i$-th cross-correlation window[18], resulting in pseudo-instantaneous Sauter diameter time series. Figure 2b shows the lower quartile ($Q_1$), median, mean and upper quartile ($Q_3$) for the measured Sauter diameters for a time series at a given measurement location. The mean Sauter diameter was the smallest close to the chute invert and increased with increasing elevation above the invert, spanning an overall range of $1$ mm $\lesssim d \lesssim 4$ mm. The smallest measured $d$ was in the order of the needle tips' outer diameter $\Phi_o = 0.60$ mm. A significant number of particles smaller than the mean Sauter diameter was detected, as evidenced by the wide, slightly right-skewed distribution of $d$ (Fig. 2b, c). Consequently, pseudo-instantaneous diameters instead of mean or median diameters were used to assess bubble–probe interactions.

**Bubble–probe interaction**. Herein we focused on a method to correct velocity underestimations due to bubble–probe interactions in the bubbly flow region ($C < 0.3$). The interaction of a bubble with the two needle tips of a double-tip probe can be described in three stages: (i) undisturbed flow upstream of the leading tip (Fig. 3a), (ii) leading tip piercing, deformation and crawling effect (Fig. 3b), and (iii) combined leading/trailing tip piercing, deformation and crawling effect (Fig. 3c). For the following derivations, it is assumed that the probe tips are aligned

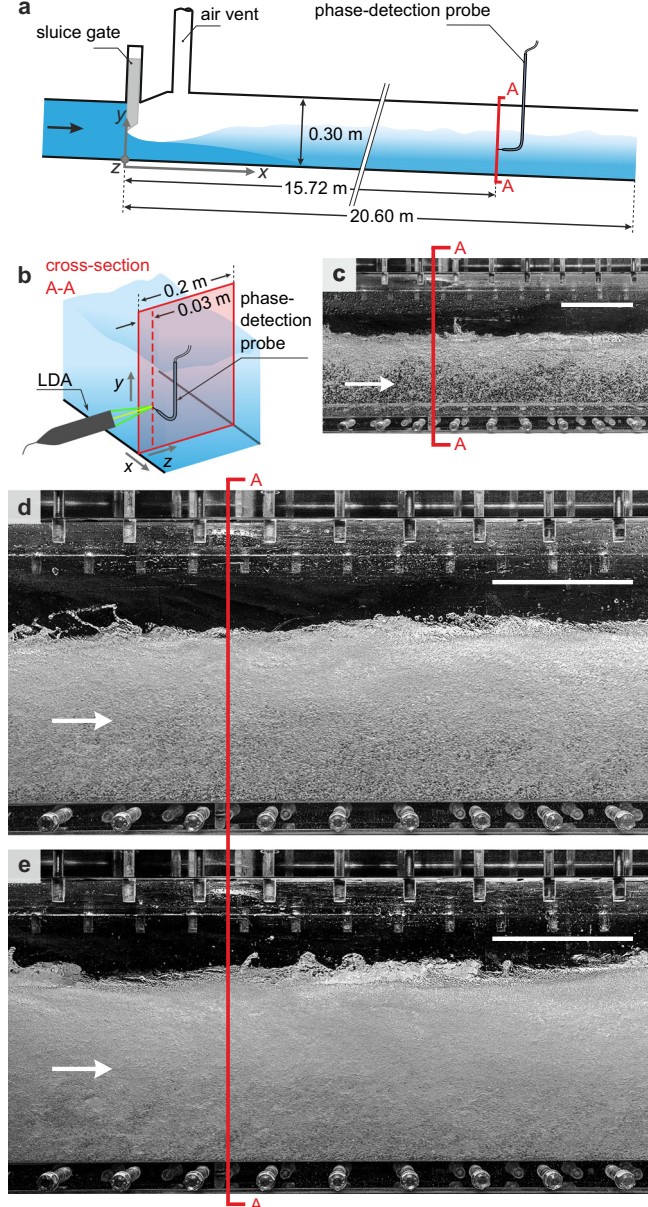

**Fig. 1 Illustration of the experimental setup and flow pattern.**
**a** Longitudinal sketch. **b** Measurement cross-section and LDA setup. High-velocity air–water flow at the measurement location for **c** Re = 0.9 · 10⁶, **d** Re = 1.3 · 10⁶, **e** Re = 1.6 · 10⁶, scale bar in (**c–e**) corresponds to 15 cm.

with the streamlines, which is a prerequisite for accurate phase-detection probe measurements[18]. Bubbles interacting with only one of the two tips are not considered in the velocity calculation[18].

The velocity underestimation due to bubble–probe interaction can be assessed by comparing the measured velocity of a pierced bubble ($u_{d,meas}$) with the real (i.e., corrected) velocity of an undisturbed bubble ($u_{d,corr}$). When a bubble is pierced by a double-tip phase-detection probe, the detection of the first and the second bubble interface by the leading and trailing tips provides travel times for each interface, $\mathcal{T}_1$ and $\mathcal{T}_2$, respectively (Fig. 3). Due to bubble–probe interactions, $\mathcal{T}_1$ and $\mathcal{T}_2$ may be different. In this case, we show that the most probable time lag from the cross-correlation analysis ($\mathcal{T}_{meas}$) tends to recover $\mathcal{T}_1$ (Supplementary Note 1) and consequently:

$$\mathcal{T}_{meas} = \frac{\Delta x}{u_{d,meas}} \approx \mathcal{T}_1 = \frac{2\Delta x}{u_{d,corr} + u_{d,1,trail}} \quad (1)$$

where $u_{d,corr}$ is the instantaneous dispersed-phase (bubble) velocity before interaction of the first interface with the leading tip, hereafter referred to as corrected dispersed-phase velocity (Fig. 3b), and $u_{d,1,trail}$ is the dispersed-phase velocity at the interaction of the first interface with the trailing tip. We assumed a linear bubble deceleration during the bubble–probe interaction based on a comparison with high-speed images from Vejražka et al.[20] (Supplementary Note 2). The undisturbed dispersed-phase velocity $u_{d,corr} = u_c - u_r$ can be estimated based on a force balance, where $u_r$ is the probe-wise bubble rise (slip) velocity.

**Force balance on a bubble.** The balance of forces on a dispersed bubble with diameter $d$, volume $V$, and cross-sectional area $A$, impacting a probe tip, is (Supplementary Note 3):

$$\overbrace{V\rho_d \frac{d\mathbf{u}_d}{dt}}^{\text{inertia}} - \overbrace{VC_{vm}\rho_c\left(\frac{D\mathbf{u}_c}{Dt} - \frac{d\mathbf{u}_d}{dt}\right)}^{\text{virtual mass}} = \overbrace{V\rho_c\frac{D\mathbf{u}_c}{Dt}}^{\text{pressure gradient}}$$

$$- \underbrace{V(\rho_c - \rho_d)\mathbf{g}}_{\text{buoyancy}} + \underbrace{\frac{1}{2}\rho_c A C_d(\mathbf{u}_c - \mathbf{u}_d)^2}_{\text{quasi-steady drag}} \quad (2)$$

$$+ \mathbf{F}_B + \mathbf{F}_{wall} - \underbrace{\mathbf{F}_\sigma - \mathbf{F}_{stag} - \mathbf{F}_{surf}}_{\text{bubble-probe interaction}}$$

where $\mathbf{F}_B$ = Basset force, $\mathbf{F}_{wall}$ = wall forces including lift and lubrication, $\mathbf{F}_\sigma$ = surface tension forces due to bubble–probe contact, $\mathbf{F}_{stag}$ = stagnation pressure force, $\mathbf{F}_{surf}$ = surface deformation force, $C_d$ = drag coefficient, $C_{vm}$ = virtual mass coefficient, $\mathbf{g}$ = gravity vector, $\mathbf{u}_c, \mathbf{u}_d$ = continuous- and dispersed-

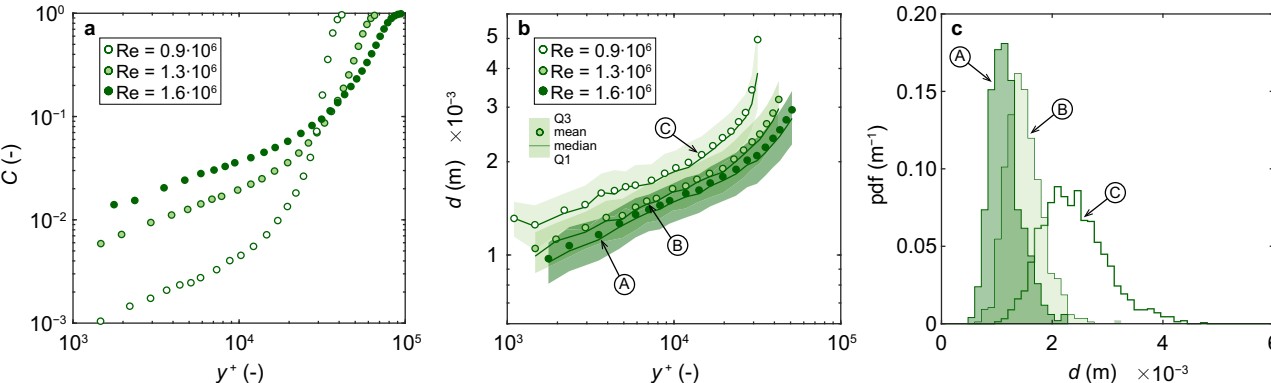

**Fig. 2 Air–water flow properties for three Reynolds numbers. a** Time-averaged void fraction $C$. **b** Sauter diameter $d$; Q1, Q3 are the lower and upper quartile, respectively. (A) to (C) indicate locations for which Sauter diameter distributions are shown in (**c**).

phase velocity vectors, $\rho_c$, $\rho_d$ = continuous- and dispersed-phase densities, respectively. $D()/Dt$ denotes the material derivative, thus $D\mathbf{u}_c/Dt$ is the total acceleration experienced by the bubble.

Vejražka et al.[20] used high-speed imaging to estimate the probe-wise surface tension force during bubble piercing as $F_\sigma = \beta \pi \sigma \Phi_o$, where $\Phi_o$ is the outer probe diameter, $\sigma$ is the surface tension coefficient and $\beta \approx 1$ for a contact line comparable to the probe's needle tip circumference. Due to the hydrophilic nature of most probe tips, the deformation of the second bubble interface (air to water) is small, resulting in an interface roughly perpendicular to the probe tip[20,27] (Fig. 3), which implies that the force component in probe-wise direction vanishes. Additional probe-wise forces are linked with the impact of the air bubble onto the needle tips[35], comprising a stagnation pressure force $F_{stag} = 1/8\ C_p\ \rho_d\ u_d^2\ \pi\ \Phi_o^2$ and a surface deformation force $F_{surf} = C_\sigma\ \sigma\ d\ We^{1/4}$ following Lebanoff and Dickerson[35], where $C_p$ and $C_\sigma$ are pressure and surface deformation coefficients, $u_d$ denotes the probe-wise component of $\mathbf{u}_d$ and the Weber number is $We = \rho_d\ u_d^2\ d/\sigma$.

In probe-wise direction, the force balance of Eq. (2) simplifies to a first-order non-linear ordinary differential equation

(Supplementary Note 3):

$$\frac{\pi d^3}{6}\left(\rho_d + C_{vm}\rho_c\right)\left(\frac{du_d}{dt}\right)$$

$$= -\frac{\pi d^3}{6}(\rho_c - \rho_d)g\cos\gamma + \frac{\pi d^2}{8}\rho_c C_d(u_c - u_d)\ |u_c - u_d| \quad (3)$$

$$- \beta\pi\sigma\Phi_o - \frac{1}{8}\ C_p\ \rho_d\ u_d^2\ \pi\ \Phi_o^2 - C_\sigma\ \sigma\ d\left(\frac{\rho_d u_d^2 d}{\sigma}\right)^{1/4}$$

where $\gamma$ is the angle between $\mathbf{g}$ and the probe-wise direction (Fig. 3) with $g = |\mathbf{g}|$. Herein, Eq. (3) was solved numerically using a Dormand–Prince Runge–Kutta method[36] (see Methods section).

**Dispersed- and continuous-phase velocities.** Uncorrected ($u_{d,meas}$) and corrected ($u_{d,corr}$) instantaneous dispersed-phase velocities were compared to continuous-phase velocities ($u_c$) obtained from LDA measurements. The velocity distributions of selected measurements showed a distinct difference between $u_{d,meas}$ and $u_c$ (Fig. 4a). The correction method shifted dispersed-phase velocities closer to the continuous-phase velocity distributions and the corrected mean (time-averaged) dispersed-phase velocities ($U_{d,corr}$) were within ±5% of the mean continuous-phase velocities ($U_c$) for all measurements. This deviation was smaller than the measurement uncertainty and comparable to the expected drift velocities between the continuous and dispersed phase of ≲1% given the large probe-wise flow velocities in our experiments (Fig. 4b). For comparison, the uncorrected mean values ($U_{d,meas}$) showed dispersed-phase velocity under-estimations of up to ≈15%, especially for small $d$. The results in Fig. 4 were consistent for repeated experiments with different phase-detection intrusive probes (Supplementary Fig. 1). The resulting turbulence levels are compared in Supplementary Fig. 2.

The time-averaged velocity profiles are shown in Fig. 4c, using inner scaling ($u^+ = U/u^*$), where all velocities are normalized with the shear velocity $u^*$ obtained from LDA measurements using the Clauser[37] method. The uncorrected dispersed-phase velocities appeared tilted compared to the log-law (Eq. (4)) obtained for the LDA data. After correcting the bubble–probe interaction bias, the dispersed-phase velocities were similar to the log-law in the inner region up to $y^+ \approx 10^4$. The deviations in the

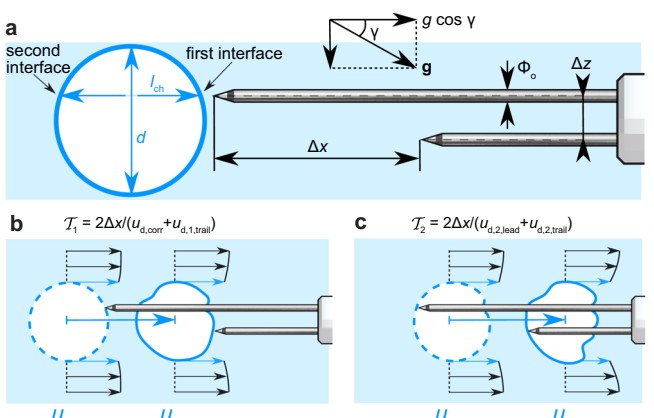

**Fig. 3 Bubble–probe interaction phases. a** Undisturbed bubble approaches a double-tip phase-detection probe. **b** Piercing and travel time of the first interface. **c** Piercing and travel time of the second interface.

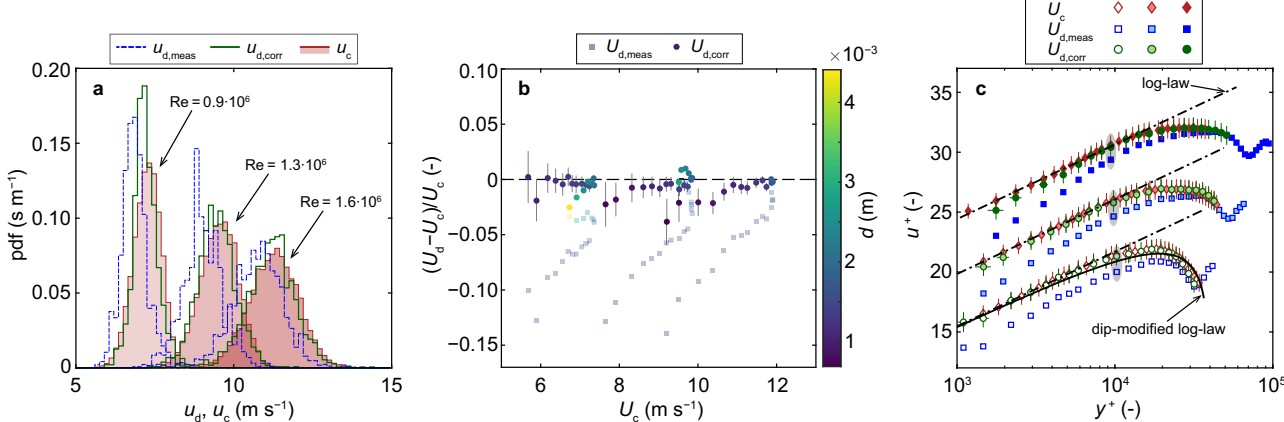

**Fig. 4 Comparison of continuous-phase velocity $u_c$ measured with LDA to corrected and uncorrected (measured) dispersed-phase velocities $u_{d,corr}$, $u_{d,meas}$ measured with CP. a** Comparison of velocity histograms at $y^+ \approx 10^4$ for different flows. **b** Mean velocity difference between dispersed and continuous phase with and without correction of interaction bias. Error bars indicate standard error (omitted for $U_{d,meas}$ for clarity). **c** Mean velocity profiles in inner scaling $u^+ = U/u^*$ compared to the log-law and dip-modified log-law[38]. Data for $Re = 0.9 \cdot 10^6$ shifted by $\Delta u^+ = -5$ and for $Re = 1.6 \cdot 10^6$ by $\Delta u^+ = +5$. All mean velocities were scaled with $u^*$ obtained from LDA measurements using the Clauser method. Gray ellipses highlight data shown in **a**, error bars indicate standard error (omitted for $U_{d,meas}$ for clarity).

**Table 1 Estimations of $u\star$ with different methods and instruments for different flow conditions (rounded to two decimals).**

| Instrument | LDA ($U_c$) | LDA ($U_c$) | LDA ($U_c$) | CP ($U_{d,meas}$) | CP ($U_{d,corr}$) |
|---|---|---|---|---|---|
| **Method** | (i) | (ii) | (iii) | (i) | (i) |
| **Re (-)** | $u\star$ (m s$^{-1}$) | $u\star$ (m s$^{-1}$) | $u\star$ (m s$^{-1}$) | $u\star$ (m s$^{-1}$) | $u\star$ (m s$^{-1}$) |
| $0.9 \cdot 10^6$ | 0.27 | 0.25 (−7.7%) | 0.25 (−9.8%) | 0.25 (−7.8%) | 0.27 (−0.4%) |
| $1.3 \cdot 10^6$ | 0.37 | 0.33 (−11%) | 0.30 (−18%) | 0.34 (−7.1%) | 0.36 (−0.8%) |
| $1.6 \cdot 10^6$ | 0.44 | 0.36 (−18%) | 0.34 (−24%) | 0.41 (−6.2%) | 0.43 (−1.2%) |

In brackets, percentage difference to Clauser method estimation for LDA data.
CP phase-detection conductivity probe.
(i) = velocity profile best-fit after Clauser[37].
(ii) = Reynolds shear stress best-fit[40].
(iii) = momentum integral-based approach[41].

outer region were primarily due to velocity dip effects, as suggested by the good agreement with the dip-modified log-law of Yang et al.[38] (Fig. 4c).

**Shear velocities**. The shear velocity $u*$ is the key velocity scale in boundary layer flows[39]. Herein, we estimated $u*$ with different methods, comprising (i) the best-fit of measured velocity distribution with the log-law after Clauser[37], (ii) the best-fit of measured Reynolds stress distribution[40], and (iii) a 2D momentum integral-based approach after Mehdi et al.[41].

Method (i) only requires mean velocity measurements in the main flow direction but relies on a priori assumptions of the parameters $\kappa$ and $y_0$[42] (see Methods section). Methods (ii) and (iii) require instantaneous 2D velocity measurement and are thus not applicable to the double-tip phase-detection intrusive probes.

For the continuous-phase velocity measurements with LDA, shear velocities from method (i) were larger than estimations using methods (ii) and (iii) (Table 1). For the two smaller investigated flow rates (Re = $0.9 \cdot 10^6$ and $1.3 \cdot 10^6$), the differences were in agreement with previous results[43–45], while differences appeared to increase with increasing flow rate and aeration. Possible reasons for this may be that 3D effects such as the velocity dip became stronger with increasing mixture flow depth, or that $\kappa$ and/or $y_0$ changed with increasing flow aeration and Re, respectively. In addition, the location of the measurements within the sidewall boundary layer ($z = 0.03$ m) may have affected the distribution of stresses, leading to differences in $u*$. The corrected dispersed-phase shear velocity estimates were within ±1.2% of the LDA results for method (i), which confirmed the suitability of the developed correction method (Table 1).

**Corrected bubble velocities: estimated deviations**. To provide a general estimate of velocity errors due to bubble–probe interactions, we applied the velocity correction method to different gas–liquid flows by numerically solving Eq. (3) for a wide range of bubble diameters, instantaneous velocities, typical probe geometries and different flow configurations (Fig. 5). Herein, $\gamma = 90°$ characterizes horizontal flows that may occur in open channels, water conveyance structures and around ship hulls, $\gamma = 0°$ represents vertical downward flows, which are often found in plunging jets, breaking waves and drop shafts, and $\gamma = 180°$ describes vertical upward flows that may arise in pipes or reactor bundles.

For each flow condition, three common phase-detection intrusive probe geometries were tested, thereby providing velocity errors that can be considered representative of previous experiments. Figure 5 illustrates that the bubble–probe interaction may lead to a velocity underestimation of up to ≈20% for typical horizontal gas–liquid flows. The underestimation was

more significant for laboratory-scale observations (i.e., smaller velocities) and for large probe tip diameters. Note that even a small underestimation (e.g. ≈ 5–10%) may be misinterpreted as phase-slip or turbulence modulation. In horizontal flows, only bubble–probe interaction forces are acting against the probe-wise direction. However, for vertical downward flows ($\gamma = 0°$), buoyancy additionally acts against the probe-wise direction, especially for large bubble diameters, thereby slightly increasing the velocity underestimation compared to $\gamma = 90°$. In vertical upward flows ($\gamma = 180°$), buoyancy acts in the probe-wise direction, thus leading to a smaller velocity underestimation compared to $\gamma = 90°$. This effect was most prominent for larger bubbles, where the bubble–probe interaction was dominated by buoyancy (gravity) effects. The bubble–probe interaction bias may still be significant for vertical upward flows, as expected velocities in these systems are typically smaller. Note that the estimated velocity bias is subject to the assumptions discussed in Supplementary Notes 2–4. Nevertheless, our results highlight that the overall magnitude of velocity bias due to bubble–probe interaction may be non-negligible in many gas–liquid flow applications.

Small bubbles moving at low velocities may be repelled by surface tension at the leading or trailing needle tip, resulting in undetected, herein called ghost bubbles. In Fig. 5, non-detected bubbles are represented by the white section in each sub-figure. Note that ghost bubbles may be pierced by the leading tip but come to a halt and drift off the probe needle before they are detected by the trailing tip. In the limit of $u_{d,1,trail} = 0$ the measured bubble velocity becomes $u_{d,meas} \approx (u_{d,corr} + 0)/2 = 0.5\ u_{d,corr}$. These ghost bubbles are due to (i) surface tension effects during bubble–probe interaction and (ii) additional buoyancy forces acting against the probe-wise direction, depending on the probe-gravity angle. These effects further extend the blind region of the probe beyond the apparent limitation of the probe dimension ($\Phi_o \gtrsim d$), which is especially relevant for laboratory-scale measurements (Fig. 5). In this context, the outer diameter of the probe tip is more relevant than the inner diameter, highlighting the advantage of finer outer probe tip diameters, which are typical for fiber-optical probes.

**Discussion**

Our proposed method accounts for bubble–probe interactions using a simplified force balance on a bubble impacting a probe (Eq. (3) and Supplementary Notes 1, 3). The solution of the force balance involves semi-empirical parameters such as the calibrated surface deformation coefficient $C_\sigma$, as well as drag and virtual mass coefficients, for which widely accepted relationships exist[46,47]. The uncertainty introduced by these coefficients in terms of a 95% confidence bound for the predicted velocity is typically below ±5 to 7% for $u_{d,corr} \gtrsim 1$ m s$^{-1}$ (Supplementary

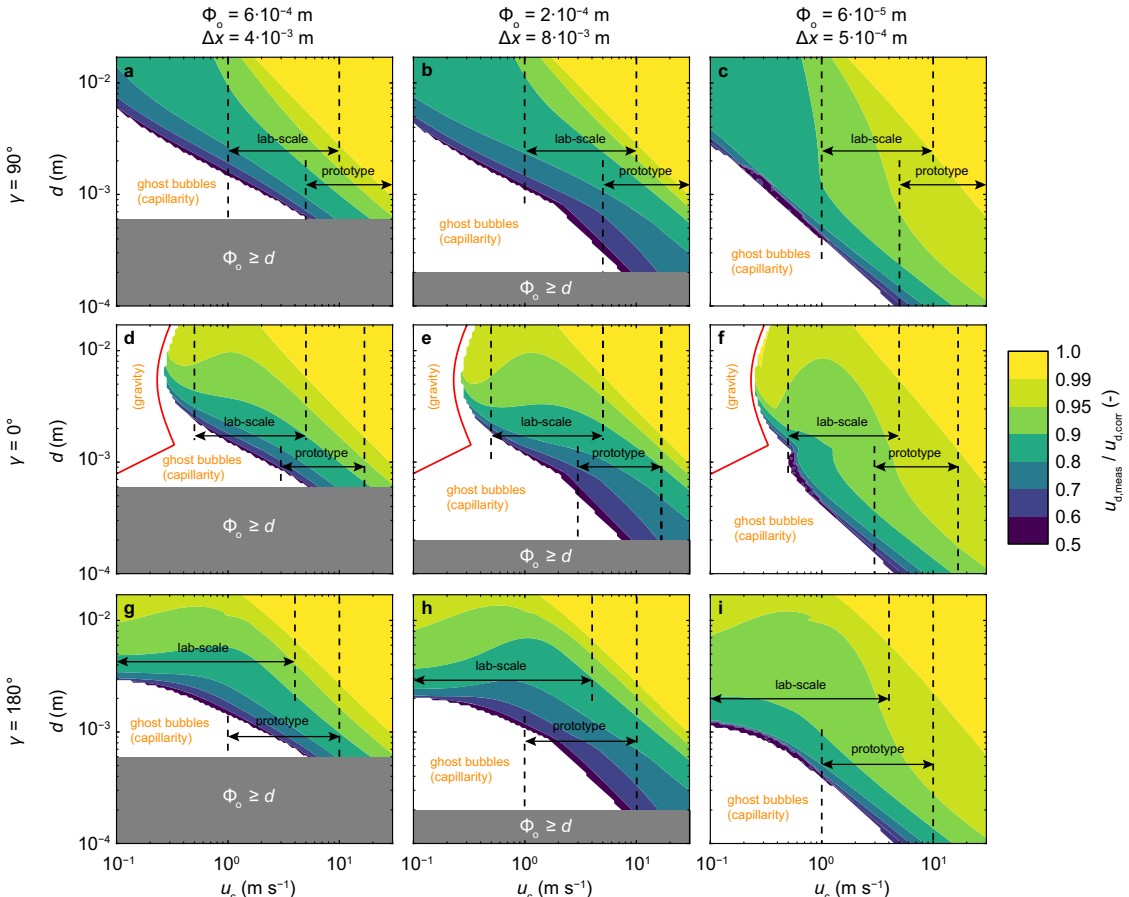

**Fig. 5 Estimated velocity underestimation due to bubble–probe interaction for gas–liquid flows depending on the Sauter diameter and the instantaneous continuous-phase velocity.** Top row corresponds to horizontal ($\gamma = 90°$), the middle row to vertical downwards flows ($\gamma = 0°$) and the bottom row to vertical upwards flows ($\gamma = 180°$). Columns compare different probe geometries. **a**, **d**, **g** conductivity probe designed to resist high velocities (this study). **b**, **e**, **h** conductivity probe designed for small to intermediate velocities[65]. **c**, **f**, **i** fine fiber-optical probe for flows around ship hulls[66]. Typical ranges for $u_c$ are indicated for comparison. The white areas represent $u_{d,1,\text{trail}} < 0$, the solid red line in (**d**–**f**) indicates the bubble rise velocity.

Note 4), which is smaller than the estimated interaction bias for typical applications and larger probes ($\Phi_o > 10^{-4}$, Fig. 5). The uncertainty may increase for smaller bubble velocities, in which case the experimental conditions need to be controlled more carefully.

We applied our method to near-horizontal high-velocity flows with negligible slip as well as to vertically rising bubbles in stagnant liquid (Supplementary Note 2), resulting in bubble diameters between 1 and 3 mm and velocities from 0.2 to 12 m s⁻¹. The corrected dispersed-phase velocities showed less than ±5% mean deviation from continuous-phase velocities, measured non-intrusively with a laser Doppler anemometer. As phase-slip was negligible, the corrected dispersed-phase velocities allowed to estimate shear velocities from phase-detection intrusive probe measurements. We showed that our method can be applied to common flow conditions and probe geometries and although we focused on double-tip probes, the developed methodology can be extended to other systems such as single-tip optical probes[48] or four-tip conductivity probes[22].

We have demonstrated that the interaction of air bubbles with the needle tips of phase-detection intrusive probes may have led to a significant underestimation of dispersed-phase velocities in previous studies of gas–liquid flows (Fig. 5). The proposed correction method allows to account for bubble–probe interactions, thereby improving the accuracy of dispersed-phase velocity measurements with intrusive phase-detection probes. Regions most likely affected by bubble–probe interactions are typically close to solid boundaries with small bubbles, low velocities and most intense transverse velocity fluctuations. This coincides with regions where the largest scatter of velocity data has typically been observed[18]. More accurate velocity measurements in near-wall regions enable estimations of shear velocities in no-slip conditions, which may allow uncovering universal properties of gas–liquid boundary layers in future studies. More accurate velocity measurements directly transfer to improved estimates of bubble size and interfacial area concentration distributions. This will enable a better description of gas–liquid flows, which are of key interest in many natural and engineering systems.

## Methods

**Experimental channel.** The model chute was located at ETH Zurich and had a usable length of 20.6 m, a width of 0.2 m, a height of 0.3 m and a bottom slope of 4% (Fig. 1). Two frequency-controlled pumps maintained the water discharge, which was controlled using a sharp-edged upstream sluice gate. An air vent provided stable, co-current stratified air–water flows and the air was also allowed to exit/enter from the uncontrolled downstream end of the model chute.

Dispersed-phase velocities were measured using two types of double-tip phase-detection intrusive probes comprising conductivity probes (CP) and fiber-optical (FO) probes. Continuous-phase velocities were measured with a laser Doppler anemometer (LDA) and a Pitot tube for comparison. All data were in close agreement, providing a validation of dispersed-phase velocity measurements with state-of-the-art continuous-phase velocity instrumentation (Supplementary Fig. 1).

**Phase-detection intrusive probes.** Table 2 shows characteristic dimensions of the deployed conductivity and fiber-optical probes, including probe-wise ($\Delta x$) and transverse tip separations ($\Delta z$), as well as inner ($\Phi_i$) and outer diameters ($\Phi_o$) of the

**Table 2 Characteristic dimensions of the deployed conductivity (CP) and fiber-optical (FO) phase-detection intrusive probes.**

| Probe | $\Delta x$ (mm) | $\Delta z$ (mm) | $\Phi_i$ (mm) | $\Phi_o$ (mm) |
|---|---|---|---|---|
| $CP_1^{(*)}$ | 4.07 | 1.02 | 0.125 | 0.60 |
| $CP_2$ | 5.36 | 1.41 | 0.125 | 0.60 |
| $FO_1$ | 5.01 | 1.00 | 0.06 | 0.20 |
| $FO_2$ | 4.98 | 1.05 | 0.06 | 0.20 |

The probe marked with $^{(*)}$ was used for the results presented in the main text, the other probes are compared in Supplementary Fig. 1.

needle tips. All probes featured a side-by-side design as recommended for high-velocity flows[28]. The conductivity probes were manufactured at the Water Research Laboratory, UNSW Sydney, and the fiber-optical probes by FiberOptics P.+P. AG, Switzerland. For each probe, the raw voltage signals of both probe tips were acquired for 300 s at 500 kHz using a NI-USB-6366 I/O unit.

Void fraction and particle frequency were calculated using a single threshold technique[11,49]. Pseudo-instantaneous interfacial velocities, mean velocities and velocity fluctuations were determined using the adaptive-window cross-correlation (AWCC) technique[17,18]. A comparison of velocities estimated with different probes is shown in Supplementary Fig. 1.

The measurement uncertainty of phase-detection intrusive probes in terms of void fraction $\sigma_C/C$ was estimated within $+0.02$ to $-0.06$[18]. Due to the dependency of $F$ on the probe's needle tip diameter, the uncertainty for $\sigma_F/F$ is unknown and not documented yet. The uncertainty of the dispersed-phase velocity $u_d$ was estimated from first principles following Johansen et al.[50], including the time resolution error, the measurement error in $\Delta x$, the convergence error and the error associated with the active probe length (i.e., length of the exposed probe tips). The last error is the most significant since it is unknown how much of the surface area of the tip must be exposed to air/water in order to trigger a phase change detection. The resulting standard uncertainty $\sigma_{u_d}$ was propagated in the calculation of all derived quantities such as mean velocity, velocity standard deviation and Sauter diameters. The deviation between the probe axis and the mean streamline was below 2°, resulting in a maximum mean velocity overestimation of <1%. Note that the instantaneous misalignment might be larger due to transverse velocity fluctuations. As this is unknown, it was not included in the error propagation described above.

**Laser Doppler anemometer (LDA)**. A Dantec Dynamics FlowLite 2D LDA was used in combination with a Dantec Burst Spectrum analyser. The LDA system consisted of two diode lasers with 200 mW power each, providing wavelengths of 532 nm and 561 nm, respectively. The LDA optic was mounted on a 3D positioning system with an accuracy of $\pm 1 \cdot 10^{-5}$ m. Samples were collected for 10 min (for $Re = 0.9 \cdot 10^6$, $Re = 1.3 \cdot 10^6$) or 15 min (for $Re = 1.6 \cdot 10^6$) or until $5 \cdot 10^5$ valid velocity bursts were recorded. The LDA system was operated in backscatter mode using high photomultiplier voltages[51,52]. As the bubbles were significantly larger than the measurement volume, predominantly instantaneous continuous-phase (water) velocities were measured by the LDA[53–55]. No further data processing was needed to account for contamination by bubble velocities.

Velocity bias effects were corrected using residence time weighting[21]. Fringe distortion and velocity gradient effects were negligible for the present test setup[56,57]. The quality of measured velocities, arrival and transit times was ensured by monotonically increasing arrival times and statistically inversely proportional measured transit times and velocities[21]. Repeated measurements for selected locations provided standard uncertainties for the mean velocity ($\sigma_{U_c}/U_c = 0.005$) and the velocity standard deviation ($\sigma_{u_{c,rms}}/u_{c,rms} = 0.008$).

**Pitot tube**. Pitot tube measurements were performed to validate the continuous-phase velocity measurements with the LDA (Supplementary Fig. 1). The Pitot tube had a circular static pressure port, an outer diameter of 6.0 mm and an inner diameter of 1.5 mm. The total and static pressures were measured at 1000 Hz using a pressure transducer with uncertainties of ±900 and ±60 Pa, respectively. All pressure tappings were flushed with water to push out entrapped air prior to measuring. The correction of MacMillan[58] was applied to account for velocity gradients, turbulence and near-wall effects[59]. In addition, the momentum exchange factor between the dispersed and continuous phase was calculated after Adorni et al.[60] and the uncertainty of $U_c$ was estimated based on the measurement uncertainty of pressure and void fraction.

**Shear velocity estimation**. The shear velocity $u^*$ is often estimated by fitting a profile function to the measured velocity profile[61]. Herein, we applied the method of Clauser[37] by fitting Eq. (4) to the logarithmic region of the velocity profile:

$$u^+ = \frac{1}{\kappa}\ln\left(\frac{y}{y_0}\right) \tag{4}$$

As the present range of $y^+ = (yu^*)/v_c$ was too small to determine $\kappa$ and $y_0$ from measurements[61], an a priori knowledge of the von Kármán constant $\kappa$ and the wall offset $y_0$ was required to apply the Clauser method[37]. We adopted $\kappa \approx 0.37$ as recommended for high Reynolds number rectangular duct flows[62]. The roughness function $y_0 = 0.11v_c/u^* + 0.0033\,k_s$ was applied with an equivalent sand roughness of $k_s = 5 \cdot 10^{-5}$ m, which we determined from single-phase LDA measurements. We checked the validity of obtained $u^*$ values by comparing against estimates from other methods (Table 1).

**Velocity bias correction method**. The correction scheme evaluated $u_{d,corr}$ on the basis of Eqs. (1) and (3). We formulated an optimization problem:

$$u_{d,corr} = \arg\min_{u_{d,corr}}\left(\frac{2\Delta x}{\mathcal{T}_{CP,meas}} - (u_{d,corr} + u_{d,1,trail})\right)^2 \tag{5}$$

where $\mathcal{T}_{CP,meas}$ is the measured travel time, deduced from a cross-correlation analysis of the phase-detection probe signal using the adaptive-window cross-correlation (AWCC) technique[17,18] and $u_{d,1,trail} = u_d(t = \mathcal{T}_1)$ was calculated from Eq. (3), subject to the initial condition $u_d(t = 0) = u_{d,corr}$ with a Dormand–Prince Runge–Kutta 4 scheme[36]. Equation (5) was solved iteratively using the Levenberg–Marquardt algorithm[63]. The bubble diameter as well as related parameters $C_d$ (Supplementary Eq. 7) and $C_\sigma$ (Supplementary Eq. 15) were updated in every iteration step until the convergence tolerance ($10^{-6}$) was reached. The procedure was repeated for every element of the pseudo-instantaneous velocity time series to obtain a corrected time series $u_{d,corr}(t)$. Note that the choice of empirical coefficients in Eq. (3) reflected our experimental conditions (re-used tap water) and resulted in a good agreement between dispersed- and continuous-phase velocities for the tested flows (Fig. 4). The coefficients may require adaptions for other systems as detailed in Supplementary Note 4.

For estimations of non-horizontal flows, we calculated the pseudo-steady relative bubble velocity due to buoyancy in probe-wise direction as:

$$u_r = \text{sign}(\cos\gamma)\sqrt{\frac{4\,d\,g|\cos\gamma|(\rho_c - \rho_d)}{3\,C_d\rho_c}} \tag{6}$$

## Data availability

Data presented in the figures are available in the Supplementary Information. Additional data are available from the corresponding author upon reasonable request.

## Code availability

The code used to process the phase-detection intrusive probe signals is available under ref. [64].

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

## Acknowledgements

We thank Dr. Sandra Orvalho, Dr. Philippe Sechet, and Dr. Alain Cartellier for providing us with detailed data obtained and partly published in Vejražka et al.[20]. Rob Jenkins (WRL, UNSW Sydney) is thanked for manufacturing the conductivity probes. The first author was supported by the Swiss Competence Center for Energy Research - Supply of Electricity (SCCER-SoE).

## Author contributions

Authors list follows the First-Last-Author-Emphasis and CRediT authorship contribution statement. B.H.: conceptualization (equal), formal analysis (lead), methodology (equal), investigation, software (equal), visualization (lead), writing—original draft. M.K.: conceptualization (supporting), methodology (equal), formal analysis (supporting), software (equal), validation (equal), writing—review and editing (equal), visualization (supporting). S.F.: conceptualization (supporting), methodology (supporting), writing—review and editing (equal). D.V.: conceptualization (equal), methodology (equal), formal analysis (supporting), validation (equal), writing—review and editing (equal), visualization (equal).

## Competing interests

The authors declare no competing interests.
