## [Peer Review File · Nature Communications]

REVIEWER COMMENTS

Reviewer #1 (Remarks to the Author):

Resolving velocity bias in intrusive gas-liquid flow measurements The authors present a physics based correction for intrusive phase-detection probes used to measure bubble diameter and velocity in multi-phase flows. I found the paper to be very well written, and the experiments and analysis to be of good quality and well documented, and the correction is well-developed, and appears to be successful in its application. However, I have some reservations about the validation of this correction which could preclude its suitability for publication in Nature Communications.

My primary concern is that the correction is only validated for conditions where the slip velocity is low and the mean streamlines are essentially horizontal. This means that the results presented are only correcting for the surface tension effects of the intrusive probe. A more thorough validation would include cases where buoyancy and drag were non-negligible contributions to the correction. Furthermore, given the reliance of the surface tension on the empirical surface tension coefficient σ , the success of this correction is therefore highly dependent on the value selected for σ . Although the authors provide information as to how they obtained other empirical coefficients (C_{vm} and C_d), they do not mention how σ was obtained and it is possible that the success of their correction will be highly dependent on the accuracy of the value of σ selected.

In addition, the section on friction velocity lacks significant impact. Primarily because the authors use the Clauser approach to find their friction velocity, which assumes the validity of the inner-scaled logarithmic velocity profile using specific coefficients, and then finds u^* by finding the value of u^* which best approximates the inner-scaled logarithmic profile ($u + v_s y^+$). The authors then use the match of the scaled and corrected $u + d,1$ profile to the logarithmic velocity as a point of discussion, which is a somewhat circular argument. The reviewer recognizes that there are few good options available to determine u^* in these types of measurements, but successful observation of inner-scaling when u^* is found using the Clauser approach should never be used to conclude that 'the corrected dispersed-phase velocities followed the universal log-law' and it is a stretch to conclude that the phase-detection intrusive probes can be used to determine u^* for conditions when slip velocities are non-zero.

Other, questions/comments that arose as I read this paper include:

1. The measurements are well within the side wall boundary layer. This is going to create threedimensionalities in the corner when $y \rightarrow 0$. Does this not affect certain assumptions used in this paper (e.g. the assumption of near-zero continuous phase velocity gradients and the validity of the log-law come to mind)
2. "the high-velocity air-water flows were gradually varied and free of inlet effects". What quantity was gradually varied?
3. The correction implicitly assumes through $u_d/u_{d,1} = \tau_0/\tau$ that the diameter of the bubble does not change as it passes from the upstream tip to the trailing tip. This assumption should be stated in the text. The value d_{32} also appears in Equations 7 and 8 also assume that d_{32} is unaffected by the presence of the probe.
4. Fig. 1: The width of the channel could be placed on the figure. Also, it was not initially clear to this reviewer that both the intrusive probes and LDA were measuring along the same vertical line.
5. Fig. 4b is not mentioned or discussed in the text.

6. The turbulent profiles of Fig. A.8 are very different between the continuous and disperse phases. If the slip velocity is near zero, how can this be the case? One would think that the larger, energy containing eddies of the continuous phase would transport the bubbles at an equivalent velocity, leading to similar velocity variances.

7. The discussion of Fig. 4c is significantly weakened by using different values of u^* for the continuous and disperse phases. As noted earlier, the determination of u^* is through a fit to the log-law, therefore it is not surprising at all that the dispersed and continuous phases agree, and that they both also agree with the log-law. 1

8. The disagreement amongst the different values of u^* can also be attributed to the values of κ used in the log-law. The value $\kappa = 0.37$ was proposed in Marusic et al. (2010) for internal channels. It's likely that an experimental configuration like the one used here is probably closer to a turbulent boundary layer, for which the value of $\kappa = 0.39$ is finding more popular use.

9. Page 5: "For the two smaller investigated flows, the differences were in agreement with previous results". What is meant by smaller? Smaller Reynolds number?

10. The discussion section mentioned the semi-empirical parameters of drag and virtual mass coefficients, but failed to mention the surface tension coefficient.

Reviewer #2 (Remarks to the Author):

The manuscript describes mitigation of bubble velocity bias due to surface tension in phase detection intrusive probes, which is of high relevance to multi-phase flow measurements. The authors suggest that the interaction between the bubble and the probe due to surface tension causes significant underestimation of bubble velocities. Although the authors claim to have derived the enhanced estimation of the bubble velocity from first principles, several assumptions have been made in the derivations. The actual accuracy of the method and the impact of the assumptions made (e.g. high Reynolds number) do not appear to have been objectively quantified, which is one of the major weaknesses of the current manuscript. For example:

- How large is the error in the approximation (4) of expression (3) and how does it vary with Reynolds number, turbulence intensity, degree of shear etc.?
- What is the impact of assuming a linear bubble deceleration during the contact times in equation 1? (line 149)
- What is the impact of surface tension on the ability of the probe to detect (pierce through) the bubble at all? This appears only to be mentioned briefly just before the discussion.
- What is the probe response effect if the bubble only covers one of the needles, but not the other one?
- And how large are both of these last two effects on the bias / measurement error?
- The continuous-phase velocities (u_c) are obtained from LDA measurements. But are these correctly measured by the used LDA processor? (line 206) See more in-depth comments regarding this below.

Lines 338-343: "Direct observations of bubble-probe interactions through high-speed imaging techniques would allow to refine the description of the interaction process, possibly including deformation effects. However, we estimate the improvement over the existing method with semi-empirical calibration parameters to be significantly smaller than the difference between the current method and uncorrected data." Has this been properly substantiated?

The bias correction methodology is presented as general between measurement techniques and flow cases, but the end results do not appear to be based on first principles. This may be a misunderstanding, but it is

unclear how the results can be generalized in this manner given the material presented in the manuscript. An exhaustive list of the implemented assumptions and a quantification of the effect of these assumptions not being valid (analytical or empirical) would significantly strengthen the manuscript.

How is the LDA measurement processed by the commercial processor employed? The high turbulence intensities near the wall makes it particularly crucial that the data is processed and analyzed correctly to avoid high velocity bias. Does the processing provide the correct values of the necessary parameters (velocity, arrival time and transit/residence time)? Unfortunately, in the experience of the referee, commercial processors often tend to produce incorrect values of the same. Please ensure validity of the data, e.g. that arrival times are monotonically increasing and that residence times and velocities are statistically inversely proportional (symmetrically for positive and negative velocities near e.g. walls).

Lines 348 – 350: “Regions most likely affected by bubble-probe interactions are typically close to solid boundaries with small bubbles, low velocities and most intense transverse velocity fluctuations.” The same near wall regions are also known to be the most difficult to measure accurately the continuous-phase velocities due to high turbulence intensities, low speeds and reflections. These regions are also some of the most critical with regards to quality of the data output by the processors. The high velocity bias is typically substantial there and errors in residence times or other output data by the processor will cause errors in any thereof derived statistics.

More detailed comments:

Line 232-235 This conclusion does not seem to be supported in the general case.

Line 259 “lead” should be “led”

Figure 4. Which measurement points are illustrated in a and b? And how large is the measuring volume compared to Δx / the LDV resolution of the measurements in figure 4c?

Table 1. The precision of the presented friction velocity is quite high (three decimals of m/s). Given the sources of bias present and not least the measurement accuracy, is this precision realistic?

Thank you for an interesting manuscript and I hope that the authors find the comments useful in improving your work.

With best regards,
Clara Velte
Department of Mechanical Engineering, Technical University of Denmark

Reviewer #3 (Remarks to the Author):

Comments on the manuscript entitled “Resolving velocity bias in intrusive gas-liquid flow measurements.” (Ref. NCOMMS-20-24917) by B. Hohermuth, M. Kramer, S. Felder & D. Valero

This paper studies the flow evolution of gas-liquid mixtures, with a focus on the measurement and evaluation of bubble velocity. A correction method is proposed to minimize the underestimation of dispersed-phase velocities by intrusive probes. The topic of the paper is of interest for the readers of Nature Communications. However, the paper is characterized by some significant weaknesses and a moderate revision, also along the lines below proposed, is needed to make the paper worthy of publication.

Specific issues

The following specific issues require attention:

1. lines 8-10, left column. Notably, such accurate velocity measurements are also essential to properly evaluate gas-liquid interaction terms in integral equations for dispersed fluids, as shown by Brocchini & Peregrine (2001). This should be properly recalled here;
2. lines 130-132, right column. Has any estimate been made of the angle between probe axis and flow direction?
3. lines 250-259, left column. The explanations here provided for differences between results by methods (i), (ii) and (iii) are fairly vague and not entirely convincing. Some better support is needed here;
4. lines 399-406, left column. More details of this uncertainty analysis are needed, in particular in relation to the convergence error;
5. lines 416-418, left column. These papers all refer to a very specific physical setting of bubble aligned (mainly by gravity, bubble column, bubble trains) and largely spaced. Given the present fairly different physical setting, doubts arise on a direct extrapolation of such results to the present conditions. This should be properly addressed;
6. lines 480-489, right column. Being this the core of the entire paper, a more detailed description of the correction method should be provided and a summary given also in the Results section, to avoid the reader go back and forth between these two sections;
7. end of page 11. Inspection of the figure suggests me that the bubble-probe interaction correction (green symbols) leads to minor improvements over the uncorrected measurements (blue symbols), both being very far from the LDA data, especially for the largest Reynolds number flows ($Re = 1.3 \times 10^6$ and 1.6×10^6). This requires some re-thinking.

References

- Brocchini, M. & Peregrine, D.H. (2001). The dynamics of strong turbulence at free surfaces. Part 2. Free-surface boundary conditions. *J. Fluid Mech.* 449, 255-290.

Eidgenössische Technische Hochschule Zürich
Swiss Federal Institute of Technology Zurich

**Laboratory of
Hydraulics, Hydrology and Glaciology
Department of
Civil, Environmental and Geomatic Engineering**

ETH Zurich
Dr. Benjamin Hohermuth
Postdoctoral Researcher
HIA C 52.2
Hönggerbergring 26
8092 Zurich, Switzerland

Phone +41 44 632 55 41
hohermuth@vaw.baug.ethz.ch
www.vaw.ethz.ch

Reviewers #1 to #3
Nature Communications
Ref. NCOMMS-20-24917

Zurich, February 27, 2021

Replies to reviewer comments

Dear Reviewers,

We would like to thank the reviewers for their insightful comments. In this document we provide a reply to each of them. Prior to that, a general remark is made to fully clarify some changes of the developed correction scheme.

First, we slightly reformulated the measured travel time (Eq. 1, Fig. 3 of the revised manuscript), which is now in direct alignment with the detection of interfaces as seen by the phase-detection probe (detailed in Appendix A.1). In this context, the new formulation is more intuitive compared to the previous version.

Second, based on the reviewers' comments, we have validated our correction scheme against a completely different flow situation (bubble column), representing a buoyancy driven flow with high slip-velocities. We have obtained this data set from Vejrazka et al. (2010), who generously shared their data from high-speed videos of bubbles impacting phase-detection intrusive probes.

Upon a detailed analysis of the data of Vejrazka et al. (2010), we realized that the correction scheme required an expansion to more accurately represent the physics of the particle-probe interaction at high velocities. Inspired by Lebanoff and Dickerson (2020), we included two additional force terms into the original force balance (stagnation pressure force and surface deformation force), which characterise the impact force of the bubble onto the needle. The derivation of the adapted force balance is detailed in Supplementary Appendix A.3, where we explain the individual terms and discuss the key assumptions of the developed correction scheme, and conduct an order of magnitude analysis that exemplifies their relevance for different Reynolds numbers and bubble diameters.

We believe that the expanded correction scheme and the uncertainty/limitations analysis conducted is more complete than in the previous manuscript. We hope that these additional efforts satisfy the reviewers' comments. Our work represents an important step in the dissection of velocity biases of phase-detection intrusive measurements, especially given that those biases have been often neglected by several research communities.

Sincerely,

Benjamin Hohermuth, Matthias Kramer,
Stefan Felder, and Daniel Valero

Laboratory of Hydraulics,
Hydrology and Glaciology

Eidgenössische Technische Hochschule Zürich
Swiss Federal Institute of Technology Zurich

**Laboratory of
Hydraulics, Hydrology and Glaciology
Department of
Civil, Environmental and Geomatic Engineering**

ETH Zurich
Dr. Benjamin Hohermuth
Postdoctoral Researcher
HIA C 52.2
Hönggerbergring 26
8092 Zurich, Switzerland

Phone +41 44 632 55 41
hohermuth@vaw.baug.ethz.ch
www.vaw.ethz.ch

Reviewer #1
Nature Communications
Ref. NCOMMS-20-24917

Zurich, February 27, 2021

Point-by-point response to Reviewer #1

Dear Reviewer #1,

We would like to thank the reviewer for the detailed review and the very useful observations and comments. We believe that these constructive comments have helped to further improve the manuscript. Please, find below a detailed point-by-point response to your comments.

The authors present a physics based correction for intrusive phase-detection probes used to measure bubble diameter and velocity in multi-phase flows. I found the paper to be very well written, and the experiments and analysis to be of good quality and well documented, and the correction is well-developed, and appears to be successful in its application. However, I have some reservations about the validation of this correction which could preclude its suitability for publication in Nature Communications. My primary concern is that the correction is only validated for conditions where the slip velocity is low and the mean streamlines are essentially horizontal. This means that the results presented are only correcting for the surface tension effects of the intrusive probe. A more thorough validation would include cases where buoyancy and drag were non-negligible contributions to the correction.

We thank the reviewer for this very valuable comment and we acknowledge the need for further validation, especially as the correction scheme aims at a wide applicability. Therefore, we have undertaken a great effort to validate the correction for a buoyancy driven flow with high slip-velocities (using the data set of Vejrazka et al. 2010), which has led to an expansion of the correction scheme (as detailed above in the "Replies to reviewer comments"). The original force balance was expanded by an impact force, comprising the stagnation pressure force and the surface deformation force (Supplementary Appendix A.3). The dissection of these force terms, as well as a detailed analysis of underlying assumptions and the uncertainty in the semi-empirical parameters, are specified in Supplementary Tables A.4 and A.5, aiming at high transparency to facilitate future applications across a wide range of flow situations. The revised correction scheme has now been validated against two completely different flow situations comprising buoyancy driven bubble column versus gravity driven high-Reynolds number free-surface flow, which cover a wide range of Reynolds and Weber numbers and we hope that this can alleviate the reviewer's concerns.

Furthermore, given the reliance of the surface tension on the empirical surface tension coefficient σ , the success of this correction is therefore highly dependent on the value selected for σ . Although the authors provide information as to how they obtained other empirical coefficients (C_{vm} and C_d), they do not mention how σ was obtained and it is possible that the success of their correction will be highly dependent on the accuracy of the value of σ selected.

We thank the reviewer for this comment. We agree with the reviewer that a more thorough documentation of the effects of key coefficients (including the surface tension) was needed. We have now performed a detailed

Point-by-point response to Reviewer #1

sensitivity analysis to document the effects of drag coefficient, virtual mass coefficient, and surface tension coefficients on the developed correction scheme, see new Supplementary Appendix A.6. In our sensitivity analysis for all coefficients we used accepted ranges from the literature (Supplementary Table A.5). Note that we calibrated the surface deformation coefficient due to missing literature data on this. Considering a range of combinations of these coefficients, our sensitivity analysis showed that the 95% confidence interval for velocities > 1 m/s are below ± 5 to 7% (Supplementary Figure A.13). For smaller velocities the confidence bound increases, indicating that the experimental conditions including water quality (e.g. surfactants, particulate matter, temperature) need to be controlled more carefully in these cases.

In addition, the section on friction velocity lacks significant impact. Primarily because the authors use the Clauser approach to find their friction velocity, which assumes the validity of the inner-scaled logarithmic velocity profile using specific coefficients, and then finds u^* by finding the value of u^* which best approximates the inner-scaled logarithmic profile (u^+ vs y^+). **The authors then use the match of the scaled and corrected $u_{d,1}^+$ profile to the logarithmic velocity as a point of discussion, which is a somewhat circular argument.**

We thank the reviewer for this important comment. We agree with the reviewer's observation that this requires clarification and correction in the manuscript. First, we would like to stress that the presentation of the shear velocity estimation is only a side product of our main contribution, which is the estimation of velocity bias in intrusive air-water flows. Applying our new approach, we were able to correct the mean velocities measured with intrusive phase-detection probes. The corrected mean velocities matched the LDA data well, as shown in the non-dimensional Figure 4c, while the uncorrected velocities of the intrusive phase-detection probe underestimated the velocities in particular in the region close to the bed (Figure 4c). The close matching of the velocities of the two instruments allowed us to also compare the shear velocities estimated from the mean velocity profiles using the Clauser method.

Many approaches exist to estimate shear velocities. However, there are limited reliable approaches to estimate the shear velocity based upon mean velocity profiles. Since the phase-detection probe only provided the mean velocities, we used the same approach (Clauser method) for both instruments.

We would like to point out that the Clauser method, despite being based on fairly strong assumptions, is still widely used in turbulent boundary layer research (e.g. Connelly et al. 2006, Schultz and Flack 2007, Monty et al. 2009, Volino and Schultz 2018, Peruzzi et al. 2020). Volino and Schultz (2018) state that for "*fully developed boundary layers, the Clauser fit remains a simple and preferred method for most zero pressure gradient cases*", while Connelly et al. (2006) compared the Clauser method with measurements of the velocity gradient in the linear viscous sublayer, finding uncertainties of $\pm 3\%$ for smooth walls and $\pm 5\%$ for rough walls.

Using the Clauser method we identified similar shear velocities for the two instruments as shown in the updated Table 1. The LDA data also allowed the computation of the shear velocities using the Reynolds stresses and a 2D momentum integral-based approach. The shear velocities are listed in Table 1 including their differences with the Clauser method for the LDA. For more clarity we have reordered Table 1.

Lastly, we would also like to clarify that we used different u^* (based on the Clauser method) in the original Fig. 4c. However, the differences in u^* were small (see Table 1), leading to indistinguishable differences of u^+ when compared to using the same u^* for all data. In the revised manuscript we have used the same shear velocity (estimated with the Clauser method for LDA data). Using this normalisation allowed us to remove any circular argument. We thank the reviewer for his/her comment, which allowed us to more precisely define the shear velocity and to avoid confusion. We have also clarified this in the text (lines 200 - 258).

- Connelly, J. S., Schultz, M. P. and Flack, K. A. (2006). Velocity-defect scaling for turbulent boundary layers with a range of relative roughness. *Experiments in Fluids* 40, 188–195.
- Monty, J. P., Hutchins, N., Ng, H. C. H., Marusic, I., and Chong, M. S. (2009). A comparison of turbulent

pipe, channel and boundary layer flows, *Journal of Fluid Mechanics* 632, 431–442.

- Peruzzi, C., Poggi, D., Ridolfi, L. and Manes, C (2020). On the scaling of large-scale structures in smooth-bed turbulent open-channel flows, *Journal of Fluid Mechanics* 889.
- Schultz, M. P. and Flack, K. A (2007). The rough-wall turbulent boundary layer from the hydraulically smooth to the fully rough regime, *Journal of Fluid Mechanics* 580, 381–405.
- Volino, R. J. and Schultz, M. P. (2018). Determination of wall shear stress from mean velocity and Reynolds shear stress profiles, *Physical Review Fluids* 3, 034606.

The reviewer recognizes that there are few good options available to determine u^* in these types of measurements, but successful observation of inner-scaling when u^* is found using the Clauser approach should never be used to conclude that ‘the corrected dispersed-phase velocities followed the universal log-law’ and **it is a stretch to conclude that the phase-detection intrusive probes can be used to determine u^* for conditions when slip velocities are non-zero.**

We thank the reviewer for this important comment. As pointed out in our previous response, we agree that we cannot conclude that there is a universal matching of the air-water mean velocity profiles with the log-law using u^* determined with the Clauser method and thus we have removed this statement from our manuscript. However, as pointed out above, this is not the main focus of the manuscript, which is the correction of the mean velocities in air-water flows. The present study revealed a close agreement between corrected air-water flow velocities and LDA measured mean velocities. This is significant since it suggests that the intrusive phase-detection probes should be able to measure the mean velocities accurately in open channel flows, including in regions unaffected by the side-wall. We have clarified the document text to better highlight the purpose of the present study.

Overall, we recognize the limitations of the Clauser method and hope to have made its limitations more clear in the revised text. We further acknowledge that phase-detection intrusive probes can only be used for an approximation of u^* if the slip velocity is negligible and adapted the main text accordingly.

1. The measurements are well within the side wall boundary layer. This is going to create three-dimensionalities in the corner when $y \rightarrow 0$. Does this not affect certain assumptions used in this paper (e.g. the assumption of near-zero continuous phase velocity gradients and the validity of the log-law come to mind)

We thank the reviewer for this important comment. We agree that the presence of the side-wall has an effect on the velocity distributions including the log law. As pointed out above, we removed any statement hinting that our data support the universal log-law (using $0.37 < \kappa < 0.41$) for aerated flows, given the limitations raised by the reviewer. As explained above, the primary purpose of this manuscript is the assessment of the velocity bias in intrusive air-water flow measurements and not the determination of the universal flow properties. Our results show the close agreement in mean velocities (in absolute terms) using the correction scheme. This is a remarkable improvement on previously reported velocity data. To acknowledge the influence of the side-wall, we added the following statement in lines 253-255 in the revised manuscript: “In addition, the location of the measurements within the side wall boundary layer ($z = 0.03$ m) may affect the distribution of stresses, leading to differences in u^* .”

Concerning the relaxation of the hypothesis of near-null continuous phase velocity gradients, we agree that those may become more important near the side-wall than in the center of the flume. We have added extra text in Supplementary Appendix A.3 to better clarify the flow conditions next to the wall: “during the bubble-probe interaction, changes in the bubble velocity u_d are larger than changes in the carrier flow u_c , and happen faster than changes in the undisturbed flow ($t_p/T \ll 1$, with t_p the particle timescale and T the turbulent integral timescale); velocity changes due to bubble-probe interaction happen at a time scale $t_p = (d + \Delta x)/u_d$, while the turbulent integral timescale is proportional to $T \sim y/u_c$. Thus, $\mathbf{F}_{grad} \approx 0$ during the particle-probe interaction as far as $(d + \Delta x)/u_d \ll y/u_c$, which holds valid within the bubbly flow region $C < 0.3$ for $y < (d + \Delta x)$ ”

Note that the last condition is satisfied for all our measurement conditions, also if applied to $z < (d + \Delta x)$. We believe that this correction scheme will also hold in areas less affected by side-wall effects enabling future non-biased research of velocity properties in air-water flows.

2. the high-velocity air-water flows were gradually varied and free of inlet effects". What quantity was gradually varied?
In open-channel hydraulics, gradually varied flow describes flows where the rate of variation of flow depth with respect to distance is small. Consequently, the pressure distribution is close to hydrostatic and the streamlines are almost parallel. We adjusted the wording and included a key reference on open channel flow (line 101).
3. The correction implicitly assumes through $u_d/u_{d,1} = T_0/T$ that the diameter of the bubble does not change as it passes from the upstream tip to the trailing tip. This assumption should be stated in the text. The value d_{32} also appears in Equations 7 and 8 also assume that d_{32} is unaffected by the presence of the probe. The reviewer is correct in stating that we neglect bubble deformation effect caused by the interaction with the probe as the undisturbed bubble diameter d for the calculation of gravity force (F_g), pressure force (F_p), drag force (F_d) and virtual mass force (F_{vm}), see Supplementary Eq. (A.14). Note that the bubble size d is updated in every iteration loop of the correction scheme to account for the fact that the original estimate of d is too large because $u_{d,corr} > u_{d,meas}$, see section "Velocity bias correction method". We would like to add that the new force terms F_{surf} (Supplementary Eq. A.13) accounts for forces due to surface deformation of the bubble upon impact on the needle tip. We would like to stress that this is not in contradiction with the use of d in Supplementary Eq. A.14, which is because (i) the equivalent diameter of a deformed bubble may still be similar compared to its undisturbed state and (ii) the surface deformation becomes dominant at larger R -numbers, which implies that the assumption of a constant d in the other forces does not have a large influence. We have added these thoughts to the revised version of the manuscript, see Supplementary Appendix A.3.
4. Fig. 1: The width of the channel could be placed on the figure. Also, it was not initially clear to this reviewer that both the intrusive probes and LDA were measuring along the same vertical line.
We thank the reviewer for this comment and have adapted Fig. 1 accordingly.
5. Fig. 4b is not mentioned or discussed in the text.
We thank the reviewer for this comment. We have noticed that Fig. 4b should have been indicated in line 228 of the original manuscript, which has now been corrected.
6. The turbulent profiles of Fig. A.8 are very different between the continuous and disperse phases. If the slip velocity is near zero, how can this be the case? One would think that the larger, energy containing eddies of the continuous phase would transport the bubbles at an equivalent velocity, leading to similar velocity variances.
We thank the reviewer for this comment. First, we would like to emphasize that time-averaged slip velocities in high-velocity air water flows, as represented by our tunnel chute experiments, are oftentimes assumed to be negligible (Rao and Kobus 1971, Cain and Wood 1981, Chanson 1996), as explicitly stated in Chanson (2013, p. 232): "a number of field and laboratory data sets demonstrated that the high-velocity gas-liquid flows behave as a quasi-homogenous mixture and the two phases travel with a nearly identical velocity, i.e. the [time-averaged] slip velocity is negligible."
However, instantaneous slip velocities may be non-zero and the comment of the reviewer is therefore valid. There may be two main reasons for the observed difference in turbulence level, including: (i) instrumentation effects and/or (ii) turbulence modulation. At the current stage, we cannot differentiate between these effects due to limitations in the design of phase-detection intrusive probes, but we would like to elaborate in the following.

In terms of measurement effects, we would like to point out that the accurate measurement of instantaneous velocities with a dual-tip phase-detection probe is challenging, especially in highly aerated flows. We recently introduced a novel signal processing technique (AWCC = adaptive window

cross-correlation), which has led to a drastic reduction of the measurement error in turbulence levels (more than one order of magnitude reduction in error), revealing a significantly different distribution shape of turbulence levels when compared to previous observations (as demonstrated in Fig. 1b and Tables 1,2 in Kramer et al. 2021). We therefore believe that the close agreement between LDA data and phase-detection turbulence measurements represents a step forward in the research of high-velocity air-water flows. Despite these advances, we cannot completely rule out velocity biases due to non-streamline alignment with the probe tips (Kramer et al. 2021), which is due to vertical and transverse velocity fluctuations. In the present study, the measurements were conducted relatively close to the wall and hence transverse velocity fluctuations may have affected the calculated streamwise turbulence levels of the intrusive phase-detection probe. To be fully transparent, we have added the following statement to the revised manuscript (Supplementary Appendix A.5): *“Applying the correction scheme reduced the turbulence intensities, especially for the largest flow rate, while differences in velocity fluctuations between the continuous and the dispersed phases were still present. These differences may be caused by (i) instrumentation effects and/or (ii) turbulence modulation. At the current stage, we cannot differentiate between those mechanisms due to limitations in phase-detection intrusive probes. Further research is needed to understand to what extent the larger observed dispersed-phase turbulence intensities are caused by instrumentation or 3D flow effects.”*

- Cain, P. and Wood, I. R. (1981). Measurements of self-aerated flow on a spillway. *J. Hydraulic Div.* 107(HY11), 1425–1444.
- Chanson, H. (2013). Hydraulics of aerated flows: qui pro quo? *Journal of Hydraulic Research* 53(3), 223–243
- Kramer, M., Hohermuth, B., Valero, D., Felder, S. (2021). On velocity estimations in highly aerated flows with dual-tip phase-detection probes - closure. *International Journal of Multiphase Flow*, 103475, <https://doi.org/10.1016/j.ijmultiphaseflow.2020.103475>.
- Rao, N.S.L. and Kobus, H.E. (1971). Characteristics of self-aerated free-surface flows. *Water and waste water/current research and practice* 10, Eric Schmidt Verlag, Berlin, Germany.

7. The discussion of Fig. 4c is significantly weakened by using different values of u^* for the continuous and disperse phases. As noted earlier, the determination of u^* is through a fit to the log-law, therefore it is not surprising at all that the dispersed and continuous phases agree, and that they both also agree with the log-law.

As mentioned above, all mean velocities (continuous and dispersed phase) in Fig. 4c were normalised with the (same) friction velocity u^* , obtained from LDA measurements using the Clauser approach. As discussed above, we agree with the reviewer that the text required clarification and updating in regards to the shear velocity. We based the estimation of the shear velocity on the corrected mean velocity for the intrusive signals and the mean of the LDA. For both instruments the shear velocities were similar since the mean velocity profiles matched closely. We also calculated the shear velocities for the LDA based upon two further methods which required instantaneous velocities of the LDA for comparison. Differences between the shear velocities determined with the Clauser method and the Reynolds stresses, showed that the present data may not represent a log-law, possibly due to the presence of the wall. We have clarified this in the manuscript text to not mislead the readers.

8. The disagreement amongst the different values of u^* can also be attributed to the values of κ used in the log-law. The value $\kappa = 0.37$ was proposed in Marusic et al. (2010) for internal channels. It's likely that an experimental configuration like the one used here is probably closer to a turbulent boundary layer, for which the value of $\kappa = 0.39$ is finding more popular use.

We thank the reviewer for this important comment. We have conducted a sensitivity analysis applying different values of κ . The resulting u^* increases for larger κ -values resulting in a worse agreement with the

Point-by-point response to Reviewer #1

u^* estimates from the Reynolds shear stress fit and the 2D integral momentum balance approach. Increasing κ from 0.37 to 0.39 resulted in a 5.3% increase of u^* for the intermediate tested flow rate ($R = 1.3 \cdot 10^6$). In addition to the points discussed in reply to your general comment regarding u^* , the presence of small air bubble near the wall may lead to drag reduction, which could manifest itself in a change in κ (e.g. Marie, 1987). Overall we think that $\kappa = 0.37$ is adequate for the sake of comparing LDA and CP measurements.

- Marie, J. L. (1987). A simple analytical formulation for microbubble drag reduction. *Physico Chemical Hydrodynamics*, 8(2), 213–220.

9. Page 5: For the two smaller investigated flows, the differences were in agreement with previous results". What is meant by smaller? Smaller Reynolds number?

We thank the reviewer for pointing this out. We have added the two missing Reynolds numbers to the text. This passage now reads "*For the two smaller investigated flow rates ($R = 0.9 \cdot 10^6$ and $1.3 \cdot 10^6$), the differences were in agreement with previous results ...*".

10. The discussion section mentioned the semi-empirical parameters of drag and virtual mass coefficients, but failed to mention the surface tension coefficient.

We thank the reviewer for this important comment. We have conducted a comprehensive sensitivity analysis on several semi-empirical parameters comprising the surface tension, virtual mass and drag coefficients (see new Supplementary Appendix A.6). As explained above, the results revealed that our sensitivity analysis showed that the 95% confidence interval for velocities > 1 m/s are below ± 5 to 7% (Supplementary Figure A.13). For smaller velocities the confidence bound increases, indicating that the experimental conditions including water quality (e.g. surfactants, particulate matter, temperature) need to be controlled more carefully in these cases.

Sincerely,

Benjamin Hohermuth, Matthias Kramer,
Stefan Felder, and Daniel Valero

Dr. Clara Velte
Nature Communications
Ref. NCOMMS-20-24917

Zurich, February 27, 2021

Point-by-point response to Reviewer #2, Dr. Clara Velte

Dear Dr. Clara Velte,

We would like to thank you for the thorough review of our manuscript and the constructive comments. We have provided a detailed point-by-point response to your comments below.

The manuscript describes mitigation of bubble velocity bias due to surface tension in phase detection intrusive probes, which is of high relevance to multi-phase flow measurements. The authors suggest that the interaction between the bubble and the probe due to surface tension causes significant underestimation of bubble velocities. Although the authors claim to have derived the enhanced estimation of the bubble velocity from first principles, several assumptions have been made in the derivations. The actual accuracy of the method and the impact of the assumptions made (e.g. high Reynolds number) do not appear to have been objectively quantified, which is one of the major weaknesses of the current manuscript.

We thank you for this important comment. We have put considerable effort into addressing your comments conducting (i) additional analysis using the data set of Vejrazka et al. (2010) of buoyancy driven flows in a bubble column, (ii) expanding the force balance to include all relevant forces acting on a particle when interacting with a probe tip including a discussion of the assumptions and the relevant importance of each force term (new Supplementary Appendix A.3) and (iii) a thorough uncertainty analysis of all semi-empirical coefficients (new Supplementary Appendix A.6). We hope that this extensive effort satisfies your important comment. Instead of repeating the content of the new Supplementary Appendixes, we would like to direct you to these (Appendix A.3 and A.6). We believe that our extra analysis now better addresses the results from an objective view point to make our approach as widely applicable as possible.

1. How large is the error in the approximation (4) of expression (3) and how does it vary with Reynolds number, turbulence intensity, degree of shear etc.?

We have conducted an extensive sensitivity analysis and order of magnitude analysis for all force terms (except for the $Du_c/Dt \approx 0$ assumption), parameters and a wider range of flow conditions (see new Supplementary Appendixes A.3 and A.6.) Please, see also our reply to comment 1 of reviewer #1 concerning $Du_c/Dt \approx 0$. The discussion on $Du_c/Dt \approx 0$ is linked to the point raised by reviewer #1 on continuous-phase flow velocity fluctuations (Supplementary Appendix A.5). In this study we show that u_d reductions of 5 – 20 % can be commonly expected. As this deceleration occurs over needle distances of 4-5 mm, du_d/dt is considerably larger during bubble-probe interaction than $Du_c/Dt \approx 0$; only except if the probe is a few millimeters from the wall ($y < (d + \Delta x)$). This indicates that contribution of Du_c/Dt to force terms should be an order of magnitude smaller.

A detailed summary on assumptions follow below:

- $F_{\text{grad}} \approx 0$, given that during the bubble-probe interaction, changes in the particle velocity \mathbf{u}_d are larger than in the carrier flow \mathbf{u}_c , and happen faster than changes in the undisturbed flow [Appendix A.3, lines 772 to 777].
- In the virtual mass term, we assume that $|D\mathbf{u}_c/Dt| \ll |d\mathbf{u}_d/dt|$ during the bubble-probe interaction (hence, $D\mathbf{u}_c/Dt \approx 0$); i.e., bubble deceleration is stronger than the turbulence fluctuations [Supplementary Appendix A.3, text to Supplementary Eq. A.5].
- We can consider negligible contribution from the Basset force (F_B), as concluded from the order of magnitude analysis for flows with velocities between 0.5 m/s to 12 m/s [Supplementary Appendix A.3, text to Supplementary Eqns. A.9, A.10, Supplementary Fig. A.9]. Note that in order to address this point, a thorough reanalysis of numerical data of Mei et al. (1994) has been conducted to obtain a Basset Kernel for finite Reynolds numbers [Supplementary Fig. A.8].
- The region of influence of wall forces (lift and lubrication) is limited to distances comparable to the bubble diameter ($y \approx d$), based on previous models of Zaruba et al. (2007) [Supplementary Appendix A.3, text to Supplementary Eq. A.3].

Furthermore, a summary with main simplifications has also been included (Table A.4) to improve transparency regarding limitations.

2. What is the impact of assuming a linear bubble deceleration during the contact times in equation 1? (line 149)

We thank you for this important question. To answer the effect of deceleration, we were able to source the data set of Vejrazka et al. (2010) who kindly provided us with their data set of high-speed video data of rising bubbles impacting on a phase-detection probe. We reanalysed their data, to identify the deceleration and found that the deceleration is close to linear for both their and our data sets. We have added a new Supplementary Appendix A.2 which provides further details on the temporal evolution of bubble velocities along a probe tip.

3. What is the impact of surface tension on the ability of the probe to detect (pierce through) the bubble at all? This appears only to be mentioned briefly just before the discussion.

We thank the reviewer for the comment. We would like to clarify the results of Fig. 5 further. Phase-detection intrusive probes are only able to detect bubbles when they pierce through the probe tip. The piercing process is affected by the bubble size, the bubble velocity, as well as the tip size (and shape), and fluid properties (e.g. surface tension). Considering all these parameters, bubbles below a certain size will not be pierced by the tip of the phase-detection probe under certain flow conditions (velocities). The region where this is the case, is shown as the "white area" in each of the nine sub-plots of Fig. 5. We denoted these bubbles as "ghost bubbles" since these bubbles are "invisible" to the phase-detection probe. We have better clarified this in the manuscript (lines 300-306).

4. What is the probe response effect if the bubble only covers one of the needles, but not the other one?

The reviewer is thanked for this valuable comment. We would like to point out that the signal processing of the dual-tip phase-detection probe data was performed using the AWCC technique (Kramer et al. 2019, 2020, 2021), which relies on small windows to estimate window-averaged pseudo-instantaneous dispersed phase velocities via a cross-correlation analysis. Each window contains a certain number of dispersed-phase particles, which was set to $N_p = 8$ in the present study based on the recommendation in Kramer et al. (2020). Therefore, if 8 bubbles impact the leading tip and only 7 of them impact the trailing tip, we would still expect a realistic velocity estimation although with a lower correlation peak in the cross-correlation function. However, at some point (e.g. if only 5 out of 8 bubble impacting both tips) the correlation peak will drop further and the window may not contain reliable velocity information anymore. To detect and remove non-informative and erroneous windows, we implemented three filtering routines, comprising (i) maximum cross-correlation coefficient (correlation peak), (ii) secondary peak ratio and (the global peak has to be clearly larger than local peaks marked) and (iii) a robust outlier cutoff (to remove

outliers, similar to procedures for ADV). Overall, single-tip impacts are anticipated to be filtered out by the implemented data processing and more details, including a discussion on velocity biases, are published in Kramer et al. (2020, 2021).

- Kramer, M., Hohermuth, B., Valero, D., Felder, S. (2021). On velocity estimations in highly aerated flows with dual-tip phase-detection probes - closure. *International Journal of Multiphase Flow*, 103475, <https://doi.org/10.1016/j.ijmultiphaseflow.2020.103475>.

5. And how large are both of these last two effects on the bias / measurement error?

We thank the reviewer for the comment. To assess the effect of missing small bubbles below a certain threshold, we examined the correlation of the bubble velocities and bubble diameters via their joint probability distribution in Fig. 1 below. The measured bubble velocities $u_{d,meas}$ show a weak correlation with d with a tendency of lower velocities for smaller bubbles (Fig. 1a, c). This is in agreement with smaller bubble being more affected by the interaction bias. However, the corrected bubble velocities $u_{d,corr}$ are uncorrelated with d (Fig. 1b, d). Consequently, we do not expect a strong effect on the *corrected* velocity statistics if bubbles below a certain threshold cannot be detected. Note that the measured particle size statistics are obviously affected and that the velocity statistics may also become affected if buoyancy is dominant. This is considered a limitation of phase-detection intrusive probes and not the proposed correction scheme, in fact the correction scheme is mitigating the effects on the velocity statistics.

Figure 1. Bivariate Kernel density estimates for measured bubble velocities and diameters (a, c) as well as for corrected bubble velocities and diameters (b, d) at two different distances above the invert y . Contour lines from dark to light indicate joint probability density. Data shown for $R = 1.3 \cdot 10^6$.

The effect of single particle impacts is partly addressed in question 4. The reason for single tip impacts is

either i) the bubble is too small to touch both tips (i.e. smaller than the transverse separation of the tips) or ii) the bubble is large enough but misses the trailing tip due to transverse/vertical movement induced by transverse/vertical velocity fluctuations. As explained above, some of the single tip impacts are filtered out by the AWCC routine. Because streamwise and transverse/vertical velocity fluctuations are correlated in boundary layers, not detecting a bubble with strong transverse/vertical fluctuations could theoretically lead to biased velocity statistics. However, the AWCC technique was tested in a wide range of flow conditions and was not strongly affected by different filtering cutoffs (i.e. a more loose/strict rejection of windows containing single tip impacts) (Kramer et al. 2020). Therefore, we expect that the overall effect for the conditions in our experiments is small.

6. The continuous-phase velocities (u_c) are obtained from LDA measurements. But are these correctly measured by the used LDA processor? (line 206) See more in-depth comments regarding this below. How is the LDA measurement processed by the commercial processor employed? The high turbulence intensities near the wall makes it particularly crucial that the data is processed and analyzed correctly to avoid high velocity bias. Does the processing provide the correct values of the necessary parameters (velocity, arrival time and transit/residence time)? Unfortunately, in the experience of the referee, commercial processors often tend to produce incorrect values of the same. Please ensure validity of the data, e.g. that arrival times are monotonically increasing and that residence times and velocities are statistically inversely proportional (symmetrically for positive and negative velocities near e.g. walls). We thank the reviewer for her comment and acknowledge that correct values of transit/residence time are important for bias free velocity statistics. To address this issue, we checked that the measured arrival times were *strictly* monotonically increasing. We also examined scatter plots of residence time versus velocity (see Fig. 2 below) to ensure that residence times are statistically inversely proportional to the measured velocity. As expected for correctly operating burst processors, shorter residence times were measured for higher velocities. Additionally, we could also observe that with increasing R (i.e. increasing flow aeration) longer residence times were less likely to occur, because the chance of a bubble to intercept one of the LDA beams outside of the measurement volume increased with increasing residence time. The excellent overlap of mean velocity (manuscript Fig. 4c) and turbulence intensity (Supplementary Fig. A.11) for different R indicates that the lower probability to record long residence times did not affect the measured velocity statistics. Also, the measured mean velocities compared favourably with Pitot tube measurements (Supplementary Appendix A.4, Supplementary Fig. A.10). The comparison of velocity mean and standard deviation with and without residence time weighting (i.e., weights = 1) is below 0.6% (see Table 1 below) confirming that potential inaccuracies in transit time measurements are negligible for the comparison of LDA and phase-detection probes presented herein. We included a statement in the LDA Methods section of the manuscript [lines 434-437] stating that we checked LDA data quality following Velte et al. (2014), but did not include the Fig. 2 and Table 1 because we think this is not of interest for the broad audience.

Figure 2. Assessment of LDA data quality; plot of measured transit time against continuous phase velocity for $R = 0.9 \cdot 10^6$ **a-c**, $R = 1.3 \cdot 10^6$ **d-f** and $R = 1.6 \cdot 10^6$ **g-i** at different distances above the invert.

Table 1. Percentage difference between velocity mean and standard deviation calculated using residence time weighting and no weighting (weights = 1). Negative values indicate that values from residence time weighting were smaller.

R	U_c			$u_{c,rms}$		
	$y = 0.006$ mm	$y = 0.024$ mm	$y = 0.048$ mm	$y = 0.006$ mm	$y = 0.024$ mm	$y = 0.048$ mm
$0.9 \cdot 10^6$	-0.39%	-0.16%	-0.13%	0.55%	0.32%	0.36%
$1.3 \cdot 10^6$	-0.28%	-0.11%	-0.12%	0.40%	0.26%	0.48%
$1.6 \cdot 10^6$	-0.30%	-0.13%	-0.05%	0.16%	-0.02%	0.29%

7. Lines 338-343: “Direct observations of bubble-probe interactions through high-speed imaging techniques would allow to refine the description of the interaction process, possibly including deformation effects. However, we estimate the improvement over the existing method with semi-empirical calibration parameters to be significantly smaller than the difference between the current method and uncorrected data.”

Has this been properly substantiated?

We still think that direct observations of bubble-probe interactions would be useful to further improve the description of the interaction process. For example, Fig. 6 presented in Vejrazka et al. (2010) provides detailed insight into the bubble deformation process but is limited to small velocities. Fig. 5.5 in Thorwarth (2008) (<https://core.ac.uk/download/pdf/36417331.pdf>) shows bubble-probe interactions at 2 m/s. To the best of our knowledge, no images have been published at higher velocities. Nevertheless, while the statement regarding the magnitude of potential improvements could be substantiated - e.g. through an extensive sensitivity analysis as presented in the new Supplementary A.3, A.6 - we agree that the statement may be confusing and removed it from the manuscript.

8. The bias correction methodology is presented as general between measurement techniques and flow cases, but the end results do not appear to be based on first principles. This may be a misunderstanding,

but it is unclear how the results can be generalized in this manner given the material presented in the manuscript. An exhaustive list of the implemented assumptions and a quantification of the effect of these assumptions not being valid (analytical or empirical) would significantly strengthen the manuscript.

We agree that, while the force balance analysis is comprehensive and fundamental, the correction scheme relies on several assumptions and semi-empirical parameters. We have clarified this in the text. We included an exhaustive list of assumptions and parameter uncertainties and performed a sensitivity analysis to quantify their effects in the supplementary information. Note that particle force balances always require semi-empirical parameters for closure but are nevertheless widely used and this limitation is thus not limited to our correction. Using data of a bubble column, generously shared by Vejrazka et al. (2010), we could demonstrate the correction also applies to conditions different from our experiments if the appropriate semi-empirical parameters are used. Combining our flow conditions and the bubble column of Vejrazka et al. (2010), we covered a velocity range from 0.2 to 12 m/s for conditions with and without slip showing that the correction applies over a range of typical gas-liquid flow conditions. We would like to refer the reviewer to the newly added Supplementary Appendix A.3 and A.6 on the results of the extensive sensitivity and uncertainty analyses, and the detailed justification for the assumptions.

9. Lines 348 – 350: “Regions most likely affected by bubble-probe interactions are typically close to solid boundaries with small bubbles, low velocities and most intense transverse velocity fluctuations.”
The same near wall regions are also known to be the most difficult to measure accurately the continuous-phase velocities due to high turbulence intensities, low speeds and reflections. These regions are also some of the most critical with regards to quality of the data output by the processors. The high velocity bias is typically substantial there and errors in residence times or other output data by the processor will cause errors in any thereof derived statistics.
We agree with the reviewer that the near-wall regions are notoriously difficult to measure for many measuring techniques. However, our analysis of LDA data quality at $y = 6$ mm, i.e. the second lowest measurement point for the phase-detection intrusive probes, indicated no issues (see reply above). Note that $y^+ \approx 10^3$ at this position and mean velocities of 5.9 to 9.5 m/s with turbulence intensities of $\approx 13\%$. LDA measurements are still expected to be reliable under these conditions.

More detailed comments

1. Line 232-235 This conclusion does not seem to be supported in the general case.
We acknowledge that this statement is only valid for no-slip conditions. We explicitly added this limitation there and in the discussion of the estimated shear velocities.
2. Line 259 “lead” should be “led”
We reworded the sentence. It now uses the word “leading”.
3. Figure 4. Which measurement points are illustrated in a and b? And how large is the measuring volume compared to Δx / the LDA resolution of the measurements in figure 4c?
For better clarity, we marked the data presented in Fig. 4a with shaded ellipses in Fig. 4c.
4. Table 1. The precision of the presented friction velocity is quite high (three decimals of m/s). Given the sources of bias present and not least the measurement accuracy, is this precision realistic?
We thank the reviewer for pointing this out. We updated table 2 with rounded numbers of two decimals.

Point-by-point response to Reviewer #2, Dr. Clara Velte

Sincerely,

Benjamin Hohermuth, Matthias Kramer,
Stefan Felder, and Daniel Valero

Reviewer #3
Nature Communications
Ref. NCOMMS-20-24917

Zurich, February 27, 2021

Point-by-point response to Reviewer #3

Dear Reviewer #3,

We would like to thank you for your valuable time that you dedicated to reading our manuscript and providing your expert comments. We have provided a detailed point-by-point response to your comments below.

This paper studies the flow evolution of gas-liquid mixtures, with a focus on the measurement and evaluation of bubble velocity. A correction method is proposed to minimize the underestimation of dispersed-phase velocities by intrusive probes. The topic of the paper is of interest for the readers of Nature Communications. However, the paper is characterized by some significant weaknesses and a moderate revision, also along the lines below proposed, is needed to make the paper worthy of publication.

1. lines 8-10, left column. Notably, such accurate velocity measurements are also essential to properly evaluate gas-liquid interaction terms in integral equations for dispersed liquids, as shown by Brocchini & Peregrine (2001). This should be properly recalled here;
We thank the reviewer for his/her valuable comment and added the reference Brocchini & Peregrine (2001).
2. lines 130-132, right column. Has any estimate been made of the angle between probe axis and flow direction?
In our experiments the deviation between probe axis and mean flow direction was below 2° resulting in a mean velocity overestimation of less than 1 %. Note that the instantaneous misalignment might be larger due to transverse velocity fluctuations acting on the bubble. We added an estimation of the misalignment error in the phase-detection intrusive probes Methods section (lines 413-417).
3. lines 250-259, left column. The explanations here provided for differences between results by methods (i), (ii) and (iii) are fairly vague and not entirely convincing. Some better support is needed here;
We thank the reviewer for the comment. We agree that further clarification is needed and we have expanded further on this in the reply to comments 7 and 8 of reviewer 1.
For LDA, which provided instantaneous streamwise and vertical velocity time series, we computed \$u^*\$ using three different methods comprising (i) the Reynolds shear stress best-fit, (ii) a momentum-integral based approach and (iii) the Clauser method. All these methods are well accepted in open-channel flows while the Reynolds shear stress method is regarded as the most reliable (Nezu and Nakagawa 1993). However, in contrast to the LDA, the phase-detection probe are only able to measure mean streamwise velocities. We therefore used the Clauser method as a means to estimate the shear velocity for these probes. Considering that the calculation of the shear velocity is only a site aspect of this manuscript, we computed the shear velocity for both instruments with the Clauser method. The close agreement between the (corrected) phase-detection probe velocities and the LDA data resulted in almost identical results for

the Clauser fit (see Table 1). We would also like to point out that the Clauser method, despite being based on fairly strong assumptions, is still widely used in turbulent boundary layer research (Connelly et al. 2006, Schultz and Flack 2007, Monty et al. 2009, Volino and Schultz 2018, Peruzzi et al. 2019, etc.). Volino and Schultz (2018) state that for “fully developed boundary layers, the Clauser fit remains a simple and preferred method for most zero pressure gradient cases”, while Connelly et al. (2006) compared the Clauser method with measurements of the velocity gradient in the linear viscous sublayer, finding uncertainties of $\pm 3\%$ for smooth walls and $\pm 5\%$ for rough walls. Considering that both LDA and phase-detection probes provided shear velocities with the Clauser method, we have now used this method (i) as the reference for our study. We have updated Table 1 to better highlight differences between the Clauser method and other methods for the LDA. Please note that we have used the shear velocity estimated with the Clauser method for the LDA data, as the reference value in the plot of the velocity distributions in Figure 4c to best highlight the close agreement of the corrected phase-detection probe velocities with the LDA data.

Overall, we recognize the limitations of the Clauser method and we have adapted the text accordingly. We further acknowledge that phase-detection intrusive probes can only be used for an approximation of u^* if the slip velocity is negligible and adapted the main text accordingly. At the present stage, phase-detection intrusive probes are the only means to provide mean velocities in highly-aerated flows (note that there is only a very narrow flow region where LDA and phase-detection probe can be used at the same time, i.e. where the local void fraction is below 3% and relatively close to the wall $z \approx 30\text{mm}$), while further air-water probe developments (e.g. miniature multi-tip phase-detection probes) are necessary to obtain Reynolds stresses. This implies that an approximation of the shear stress through phase-detection probes currently relies on the mean velocity profile.

- Connelly, J. S., Schultz, M. P. and Flack, K. A. (2006). Velocity-defect scaling for turbulent boundary layers with a range of relative roughness. *Experiments in Fluids* 40, 188–195.
- Monty, J. P., Hutchins, N., Ng, H. C. H., Marusic, I., and Chong, M. S. (2009). A comparison of turbulent pipe, channel and boundary layer flows, *Journal of Fluid Mechanics* 632, 431–442.
- Nezu I, Nakagawa H. (1993) *Turbulence in Open-channel Flows*. IAHR Monograph, A.A. Balkema Rotterdam.
- Peruzzi, C., Poggi, D., Ridolfi, L. and Manes, C (2020). On the scaling of large-scale structures in smooth-bed turbulent open-channel flows, *Journal of Fluid Mechanics* 889.
- Schultz, M. P. and Flack, K. A (2007). The rough-wall turbulent boundary layer from the hydraulically smooth to the fully rough regime, *Journal of Fluid Mechanics* 580, 381–405.
- Volino, R. J. and Schultz, M. P. (2018). Determination of wall shear stress from mean velocity and Reynolds shear stress profiles, *Physical Review Fluids* 3, 034606.

4. lines 399-406, left column. More details of this uncertainty analysis are needed, in particular in relation to the convergence error;

The error propagation is explained in the cited reference of Johansen et al. (2010) and follows the ASME PTC 19.1-2005 Test Uncertainty Standard. The convergence error for the mean velocity is included as the empirical random standard uncertainty (\hat{s}_u) described by:

$$\hat{s}_u = \frac{T_{(n-1),95} s_u}{\sqrt{n}} \quad (1)$$

where n = number of measurements, $s_u = \left(\sum_{i=1}^n (u - U)^2 / (n - 1)\right)^{0.5}$ is the standard deviation and $T_{(n-1),95}$ the Student's T-value for a 95% confidence interval with $(n - 1)$ degrees of freedom. We added a convergence plot for the dispersed phase velocities below, demonstrating that the convergence error for the mean velocity was below $\pm 2\%$ and below $\pm 5\%$ for the standard deviation. We feel this adds little to the manuscript and thus did not add the figure as Supplementary Appendix.

Point-by-point response to Reviewer #3

Figure 3. Convergence plot for $R = 1.3 \cdot 10^6$ of **a** mean dispersed phase velocity and **b** velocity standard deviation for different y , only every second location shown for clarity. n = number of elements in time series.

- lines 416-418, left column. These papers all refer to a very specific physical setting of bubble aligned (mainly by gravity, bubble column, bubble trains) and largely spaced. Given the present fairly different physical setting, doubts arise on a direct extrapolation of such results to the present conditions. This should be properly addressed;

We acknowledge that the cited papers refer to different flow conditions, however, the main conclusions from these works still holds; contamination of the continuous-phase velocity by bubbles is small if the bubbles are larger than the LDA measurement volume. This holds true for our study as the control volume ($\approx 10^{-11} \text{m}^3$) is smaller than the typical bubble volume ($\approx 10^{-9} \text{m}^3$). An additional effect arising in dense bubble flows is the blocking of the laser beam by bubbles outside of the control volume. This mainly reduces the data rate and leads to a lower tendency to detect tracer particles with long residence times (Fig. 2 in answer to comment 6 of reviewer # 2). A detailed analysis of the LDA data showed that this has a negligible effect on the measured velocity mean and standard deviation, see detailed reply to comment 6 of reviewer # 2. This was also confirmed by the excellent agreement of the LDA mean velocities with Pitot tube measurements in the Supplementary Appendix A.4.

- lines 480-489, right column. Being this the core of the entire paper, a more detailed description of the correction method should be provided and a summary given also in the Results section, to avoid the reader go back and forth between these two sections;

This comment has become partly obsolete as the revised correction scheme does not allow an elegant analytical solution and is solved numerically. We thus merged the former sections on "Solution of the bubble force balance" and "Velocity bias correction method" into one section and added a clear outline of the computational loop required for the correction scheme. As the revised solution procedure uses standard numerical methods, we feel no additional explanation in the main text is necessary.

- end of page 11. Inspection of the figure suggests me that the bubble-probe interaction correction (green symbols) leads to minor improvements over the uncorrected measurements (blue symbols), both being very far from the LDA data, especially for the largest Reynolds number flows ($Re = 1.3 \cdot 10^6$ and $1.6 \cdot 10^6$). This requires some re-thinking.

We would like to thank the reviewer for pointing this out. We understand that use of the same symbols as in the figures in the main text could cause some confusion. The uncorrected velocities for CP and FO probes are shown in sub-figure **a**, while the corrected velocities are displayed in sub-figure **b**. The correction scheme works equally well for all tested phase-detection intrusive probes. We adapted the Figure A.10 to improve clarity.

Point-by-point response to Reviewer #3

Sincerely,

Benjamin Hohermuth, Matthias Kramer,
Stefan Felder, and Daniel Valero

REVIEWERS' COMMENTS

Reviewer #1 (Remarks to the Author):

The authors have adequately addressed my prior comments. It is my opinion that the manuscript is now suitable for publication in Nature Communications.

Reviewer #2 (Remarks to the Author):

Thank you for your extensive revision and efforts in improving the manuscript.

Reviewer #3 (Remarks to the Author):

The paper has been appreciably improved upon revision and publication can be granted.